# Adapting the FAST-M maternal sepsis intervention for implementation in Pakistan: a qualitative exploratory study

Sheikh Irfan Ahmed  ,[1] Bakhtawar M Hanif Khowaja,[1] Rubina Barolia,[2] Raheel Sikandar,[3] Ghulam Kubra Rind,[1] Sehrish Khan,[3] Raheela Rani,[3] James Cheshire,[4] Catherine Louise Dunlop  ,[4] Arri Coomarasamy,[4] Lumaan Sheikh,[5] David Lissauer[6]

¹Obstetrics and Gynecology, The Aga Khan University, Karachi, Pakistan
²School of Nursing and Midwifery, Aga Khan University, Karachi, Pakistan
³Obstetrics and Gynecology, Liaquat University of Medical and Health Sciences, Jamshoro, Pakistan
⁴Institute of Metabolism and Systems Research, University of Birmingham, Birmingham, UK
⁵Obstetric & Gynecology, The Aga Khan University Hospital, Karachi, Pakistan
⁶Institute of Life Course and Medical Sciences, University of Liverpool, Liverpool, UK

**Correspondence to**
Dr Sheikh Irfan Ahmed;
sheikh.irfan@aku.edu

## ABSTRACT

**Objective** A maternal sepsis management bundle for resource-limited settings was developed through a synthesis of evidence and international consensus. This bundle, called 'FAST-M' consists of: Fluids, Antibiotics, Source control, assessment of the need to Transport/ Transfer to a higher level of care and ongoing Monitoring (of the mother and neonate). The study aimed to adapt the FAST-M intervention including the bundle care tools for early identification and management of maternal sepsis in a low-resource setting of Pakistan and identify potential facilitators and barriers to its implementation.

**Setting** The study was conducted at the Liaquat University of Medical and Health Sciences, which is a tertiary referral public sector hospital in Hyderabad.

**Design and participants** A qualitative exploratory study comprising key informant interviews and a focus group discussion was conducted with healthcare providers (HCPs) working in the study setting between November 2020 and January 2021, to ascertain the potential facilitators and barriers to the implementation of the FAST-M intervention. Interview guides were developed using the five domains of the Consolidated Framework for Implementation Research: intervention characteristics, outer setting, inner setting, characteristics of the individuals and process of implementation.

**Results** Four overarching themes were identified, the hindering factors for implementation of the FAST-M intervention were: (1) Challenges in existing system such as a shortage of resources and lack of quality assurance; and (2) Clinical practice variation that includes lack of sepsis guidelines and documentation; the facilitating factors identified were: (3) HCPs' perceptions about the FAST-M intervention and their positive views about its execution and (4) Development of HCPs readiness for FAST-M implementation that aided in identifying solutions to potential hindering factors at their clinical setting.

**Conclusion** The study has identified potential gaps and probable solutions to the implementation of the FAST-M intervention, with modifications for adaptation in the local context

**Trial registration number** ISRCTN17105658.

## STRENGTHS AND LIMITATIONS OF THIS STUDY

⇒ The major strength of this study is the use of Consolidated Framework for Implementation Research (CFIR) domains, which we used for the development of interview guides to gather study data.
⇒ Data were collected from multiple levels of healthcare providers (HCPs) using different methods of data collection, that is, individual interviews and focus group discussion to triangulate our study findings and establish the trustworthiness of the study.
⇒ The key informant interviews focused mainly on the doctor's perspective due to the prominent role of doctors in the study setting which limited us to gain perceptions of other HCPs.
⇒ The study focused only on the perspective of the HCPs who have experience in the management and treatment of maternal sepsis patients to know the existing sepsis guidelines of the facility and adapt the intervention based on their experiences and feedback.

## BACKGROUND

Maternal sepsis is a major contributor to maternal morbidity and mortality worldwide.[1] Maternal sepsis is a life-threatening organ dysfunction caused by a dysregulated host response due to infection during pregnancy, childbirth and in the postpartum period.[2 3]

Globally, maternal sepsis accounts for about one tenth of maternal deaths and is the third most common cause of maternal mortality.[1 4] It was estimated that each year 75 000 maternal deaths occurred in low-income and middle-income countries due to maternal sepsis and approximately 10% of maternal deaths in Africa and Asia occur due to sepsis.[4 5] The risk of death among women who develop puerperal sepsis was higher in Africa (OR=2.71), Asia (OR=1.91) and Latin America and the Caribbean (OR=2.06) than in developed countries.[5]

Led by the WHO and other partners, a global initiative was commenced in 2015, to develop strategies aimed at improving the early recognition and management of maternal sepsis.[6] Strategies to ensure early identification and treatment of sepsis have demonstrated significant improvement in outcomes in high-income adult population settings[7] and it was necessary to translate these approaches into the maternity population and make them appropriate for low-resource settings.[8] Yet, there is very limited evidence of the implementation of such approaches specific to maternity care in low-resource settings.

Thus, a maternal sepsis bundle was developed as part of this process to improve the recognition and management of maternal sepsis in a low-resource setting. A modified Delphi approach was adopted to identify components significant to treatment and monitoring in terms of clinical importance and feasibility in resource-poor settings.[6] The components selected were: Fluids, Antibiotics, Source control, assessment of the need to Transport/Transfer to a higher level of care and ongoing Monitoring (of the mother and neonate). The bundle was named 'FAST-M' as a memorable acronym for both communication and awareness-raising.[6]

Implementation of the FAST-M intervention across 15 government healthcare facilities in Malawi was found to not only be feasible but also resulted in improved clinical care,[9] demonstrating that the intervention could assist in the early identification and management of maternal sepsis in low-resource settings.[9] This is now being tested formally as part of a large cluster-randomised trial across Malawi and Uganda.

In Pakistan, complications during pregnancy and childbirth are the leading causes of death in women, accounting for 20% of all deaths of women of childbearing age.[10–12] National figures show that 15% of maternal deaths are reported due to sepsis[12] and maternal sepsis is established as the third-leading cause of maternal mortality.[13] Globally, the incidence of puerperal sepsis is 4.4%[10] whereas in Pakistan the incidence is reported to be 10%–15%.[14]

There are national sepsis guidelines for Pakistan, which are designed to aid in the identification and management of sepsis in adults in the local settings and are modelled on the Surviving Sepsis Campaign.[15] However, these are inconsistently applied and lack a comprehensive implementation approach. There is still uncertainty about how best to optimise the implementation of evidence-based practices for prevention and management of maternal sepsis in Pakistan.

The absence of routine monitoring in most public facilities in Pakistan during labour and childbirth such as not taking vital signs of women and newborns substantially increases the risk of maternal and newborn morbidity and mortality.[16] It has been evident that the quality of care is poorer in public referral facilities than in primary healthcare facilities.[17] While the FAST-M intervention when implemented in health settings of Malawi has shown improvements in vital signs recording and timely identification and management of women with maternal sepsis.[6]

It is therefore planned to adapt and implement the FAST-M intervention in Pakistan. However, we recognise that to optimise its use in the Pakistani context requires a robust process of adaptation and redesign prior to its field testing. The implementation of the FAST-M intervention will be highly context specific. Therefore, this study aims to understand the existing sepsis management practices and behaviours to adapt the FAST-M bundle care tools in the local context. In addition, it will assist in the identification of the potential facilitators and barriers to its implementation in a low-resource setting within Pakistan.

This qualitative study was conducted in preparation for the implementation of FAST-M intervention in phase II of the study. The protocol and procedures for phases I and II of this study have been described in detail elsewhere.[18] The study findings obtained in this formative research will aid in the development of feasible methods to improve the processes and implementation of the FAST-M intervention in Pakistan.

## METHODS
### Study design
Our methods, grounded in implementation science, aimed to identify the anticipated facilitators and barriers in the implementation of FAST-M intervention at the Liaquat University of Medical Health Sciences (LUMHS), Hyderabad. Implementation research aims to identify the factors that function as barriers and enablers to specific interventions.[19] As our research question is descriptive and exploratory, this formative research adopted a qualitative research design involving both focus group discussion (FGD) and key informant interviews (KIIs) and a purposive sampling approach.

FGD and KIIs were conducted with healthcare providers (HCPs) working at the study site using interview guides structured using the Consolidated Framework for Implementation Research (CFIR).[20] The aim of FGD and KIIs was to engage health practitioners, government officials, and other key stakeholders to understand the existing practices in the study setting for maternal sepsis care, identify various facilitators and barriers that may influence the implementation of the FAST-M intervention and inform the adaptation of FAST-M bundle care tools and implementation approach according to the local context. Data collection through KIIs and FGD were to ensure data triangulation through different methods ensuring credibility of the study findings. This study is being stated as per the guidance provided in consolidated criteria for reporting qualitative research (see online supplemental file 1).

### Consolidated Framework for Implementation Research
The CFIR is a 'meta-theoretical' framework that provides an overarching analysis for implementation.[20] It offers an extensive and standardised list of constructs that allow

**Table 1** CFIR domains and associated constructs

| Domains | Constructs |
| --- | --- |
| One: Intervention characteristic | Intervention Source Evidence Strength and quality Relative Advantage Adaptability Trialability Complexity Design Quality and Packaging Cost |
| Two: Outer setting | Patient Needs and Resources Cosmopolitanism Peer Pressure External Policies and Incentives |
| Three: Inner setting | Structural characteristics Networks and Communication Culture Implementation Climate Tension for change Compatibility Relative priority Organisational incentives and rewards Goals and feedback Learning climate Readiness for implementation Leadership engagement Available resources Access to knowledge and information |
| Four: Characteristics of individuals | Knowledge and Beliefs about the Intervention Self-efficacy Individual stage of change Individual identification with organisation Other personal Attributes |
| Five: process | Planning Engaging Opinion leaders Formally appointed internal implementation leaders Champions External change agents Executing Reflecting and Evaluating |

CFIR, Consolidated Framework for Implementation Research.

researchers to identify various variables that are most relevant to a particular intervention.[21] The CFIR consists of five major domains: intervention characteristics, outer setting, inner setting, characteristics of the individuals and the process of implementation. These domains are organised into 39 constructs (table 1).

CFIR has been used in various studies to inform qualitative processes across a range of complex intervention, because this flexible framework can be tailored to different settings across multiple contexts.[20 21] We; therefore, used the tailored CFIR framework to understand critical barriers and facilitators to implementation of FAST-M intervention that need to be addressed at multiple levels if the FAST-M intervention is to be successfully optimised, and adopted in healthcare practices in Pakistan.

### Study setting
LUMHS is located in Hyderabad district, Pakistan. LUHMS is 1300 bed tertiary referral public sector hospital which serves a large number of mostly underprivileged populations. The hospital offers various facilities for both in-patient and out-patient. The hospital has three obstetrics and gynaecology (OBGYN) units and provides 24 hours emergency cover to patients coming from urban and rural areas of Sindh. It manages a high volume of cases of maternal sepsis every month. The current data from the facility shows that a total of approximately 11 205 patients were admitted to OBGYN units from the period of January to August 2021, and the maternal mortality rate was recorded as 159/11205 (1.4%). Out of these 159 deaths, 45 were due to confirmed maternal sepsis (28.3%). These indicators direct that there is a need for a robust system to early detect and manage maternal sepsis cases in the hospital.

### Patient and public involvement
There was no patient or public involvement in setting the research agenda.

### Data collection methods and study participants
HCPs working at LUMHS hospital were purposively sampled for KIIs and FGD. The letters of invitation were sent to all HCPs including Doctors (residents and faculty members), staff nurses and administrators who were involved in the management and treatment of maternal sepsis patients for at least the past 6 months from the time of invitation. All the participants who were approached by the study team agreed to participate in the study. The aim of KIIs and FGD was to explore and understand the behaviour of the existing practices and guidelines used in the hospital for sepsis management, and an appropriate system for characterising intervention and its components that can make use of this understanding. KIIs with HCPs were conducted in the meeting room and faculty offices at LUMHS hospital. A FGD was conducted in the seminar room at LUMHS hospital. A trained moderator facilitated the FGD. Interviews were scheduled according to participants' preferences and were audio-recorded following consent from study participants (online supplemental file 2).

### Data collection procedure
A semistructured interview guide was developed to explore healthcare professionals' views and attitudes towards the FAST-M intervention (online supplemental file 3), with a focus on the views on the feasibility of FAST-M implementation among healthcare professionals using five major domains of CFIR: intervention characteristics, outer setting and inner setting, characteristics of the individuals and the process of implementation. The interview guides were tailored considering each category of participants. The research team reviewed the interview guide for content and flow and trialled the guide for the length of time and appropriateness of the questions. Before beginning the interview, the qualitative researchers first described the FAST-M bundle components and the patient referral pathway (online supplemental file 4) demonstrating the utilisation of FAST-M

bundle care tools. The interview guide underwent subsequent modifications and iterations based on interviews conducted.

A free flow of information was encouraged, using probes from these discussions to obtain healthcare professionals' perceptions about the adaptation and feasibility of the FAST-M intervention. Interviews were conducted face to face in Urdu and English (KIIs=16; FGD=1). The standards of precautions for control of COVID-19 infection were followed during data collection. All study participants were screened before interviews for COVID-19 infection through a series of questions regarding their symptoms. The participants were asked to wear masks at all times during interviews and discussions. The FGD was conducted in a large seminar room to maintain physical distance between participants as a precaution for control of COVID-19 infection.

Interviews and FGD were conducted by RB, SIA, BMHK and GKR, who are part of the investigating team and are trained in qualitative research. The research questions were based on FAST-M intervention characteristics, outer and inner healthcare setting, characteristics of the individuals and the process of implementation. Detailed field notes were taken during each interview to capture non-verbal language and cues. KIIs were conducted for 20–40 min; FGD was conducted for 50 min and consisted of 12 participants in a group. Data were collected using interview guides developed on five major domains of CFIR: intervention characteristics, outer setting, inner setting, characteristics of the individuals and the process of implementation. Data were collected and analysed through an iterative process. The data collected through interviews and discussion were carried out until data saturation was achieved and no new information emerged.[22] We defined saturation as the amount of data needed until nothing new information and a meaningful conclusion drawn out about the feasibility of the FAST-M intervention was apparent and redundancy was reached.

### Data analysis
Study data were analysed using a conventional qualitative content analysis approach facilitated by NVivo V.10 (QSR International) software. First, all the audio-recordings were translated and transcribed from the local language (Urdu) into English. Transcripts were read several times to develop an interpretation of the participants' views about the feasibility of FAST-M implementation. FGD and KIIs were coded as one data set. Two investigators coded a subset of transcripts independently using separate coding that was then combined to match codes, and agreement by investigators was sought on a coding framework. Codes were formulated inductively from the transcripts related to research questions and CFIR domains. Coding discrepancies were discussed and resolved to reduce researchers' biases. Codes were then analysed into categories and then the major themes based on the data findings.

| Table 2 | Study participants |
|---|---|
| Focus group discussion with HCPs | Total FGD=1; n=12 |
| Doctors (medicine); (OBGYN) | n=3, n=5 |
| Nurses (OBGYN); (labour room) | n=1, n=1 |
| Health administrators | n=2 |
| Key informant interviews | Total KIIs=16, n=16 |
| Doctors (OBGYN); (operating room); ICU | n=8, n=1, n=2 |
| Nurses (OBGYN) | n=4 |
| Health administrators | n=1 |

FGD, focus group discussion; HCPs, healthcare providers; ICU, intensive care unit; KIIs, key informant interviews; OBGYN, obstetrics and gynaecology.

The potential barriers and facilitators and modifications in the bundle care tools were identified that were discussed and reviewed by the research team. To ensure the credibility of the research, study data were triangulated by different data sources including doctors, nurses, and administrators and through different data collection methods including FGD and KIIs, to compare alternative perspectives and to assess any inconsistencies. The hospital leadership and a subgroup of clinical care providers were directly contacted and invited to attend an interactive session to hear about the findings and reflect on whether these were considered representative of their existing practices prior to modifying the bundle care tools and adapting the intervention. This respondent's validation process enhanced rigour and established conformability.[23]

## RESULTS
In this qualitative study, 1 FGD and 16 KIIs (table 2) were conducted with HCPs (doctors, nurses and health administrators), between November 2020 and January 2021 who were involved in the management and treatment of maternal sepsis patients. Tables 3 and 4 present demographics of study participants. A baseline facility audit was alongside conducted to identify the availability of resources in the facility (online supplemental file 5). The survey findings assisted the study team to plan a practical approach for the implementation of the intervention (the audit findings will be recorded elsewhere). The qualitative findings presented in this paper aided the validation of observational findings. This helped the study team to gain feedback and insights from HCPs about their existing sepsis guidelines and resource availability. Based on these findings, the bundle care tools will be modified before implementation and the feasibility assessment.

Data analysis revealed four overarching themes: (1) challenges in existing system; (2) clinical practice

**Table 3** Demographics of participants in KIIs

| KIIs | n=16 |
|---|---|
| **Job title** | |
| Faculties from obstetrics and gynaecology (OBGYN) (professor, associate and assistant professors) | 3 |
| Faculties from family medicine (professor, associate and assistant professors) | 1 |
| Registrars, residents and medical officers (OBGYN) | 5 |
| Residents and medical officers (family medicine) | 2 |
| Registered nurses | 4 |
| Administration staff | 1 |
| **Working experience in facility** | |
| >10 years | 7 |
| > 5 years | 6 |
| 1–5 years | 3 |
| **Gender** | |
| Male | 4 |
| Female | 12 |
| **Role in the hospital** | |
| Administration | 2 |
| Leadership | 3 |
| Clinical practices | 11 |

KIIs, key informant interviews.

**Table 4** Demographics of group participants

| FGD participants | n=12 |
|---|---|
| **Job title** | |
| Faculties from obstetrics and gynaecology (professor, associate and assistant professors) | 5 |
| Faculties from family medicine (professor, associate and assistant professors) | 3 |
| Registered nurses | 2 |
| Administration registrars | 2 |
| **Working experience in facility** | |
| >10 years | 5 |
| > 5 years | 5 |
| 1–5 years | 2 |
| **Gender** | |
| Male | 4 |
| Female | 8 |
| **Role in the hospital** | |
| Administration | 2 |
| Leadership | 5 |
| Clinical practices | 5 |

FGD, focus group discussion.

**Table 5** Themes and categories

| Themes | Categories |
|---|---|
| Challenges in existing system | Shortage of HCPs in the hospital |
| | Lack of adequate resources and quality assurance |
| Clinical practice variation | Sepsis guidelines and documentation |
| | Individual care practices and HCP comfort levels |
| Healthcare providers' perceptions about FAST-M | Understanding of the FAST-M bundle |
| | Perceptions about significance of FAST-M |
| | Identifying solutions to the application of FAST-M |
| Development of HCPs readiness for FAST-M implementation | Understanding and identifying gaps |
| | Consensus building for FAST-M implementation |

HCPs, healthcare providers.

variation; (3) HCPs' perceptions about FAST-M and (4) Development of HCPs readiness for FAST-M implementation. Table 5 demonstrates the identified themes and categories.

## Challenges in existing system
### Shortage of HCPs in the hospital

A majority of the study participants reported challenges in the existing sepsis management practices. The major challenge reported by HCPs is the increased volume of patients coming to the OBGYN inpatient wards and emergency room. The increased number of patients exaggerates the workload on HCPs. The issue of a high patient to doctors' ratio that is 6:1; and a high patient to nurses' ratio that is 20:1 was raised by a majority of study participants. There is a shortage of health workforce considering the influx of patients in the unit which is a hindering factor for provision of quality healthcare services.

> Being a tertiary level hospital, being a civil hospital and the main hospital, we are facing an increase patients flow on daily basis (KII- Senior Registrar- OBGYN)

> On floor, we have 6 doctors and you think how many patients are there. Sometimes we have 36 admissions; sometimes we have around 40 admissions. So, you can see for doctors to patients ratio it is around 6:1 and for staff, they are sometimes present and sometimes not (KII- Senior Registrar)

HCPs identified that there is a considerable shortage of nurses in the hospital for the care of patients. The importance of nurse's role was acknowledged by all the key informants and focus group participants, and they emphasised the shortage of nurses for sepsis

management in the hospital as a key challenge, with only one or two nurses assigned to 20 patients in each shift.

As it was stated:

Yes we are short of staff nurses. Look, if we have around 32 to 40 patients so there is only one nurse for their care or hardly two (KII- Staff Nurse)

In emergency room, we do not have staff nurses available, so the doctor is responsible for maintaining IV line and catheterization. If there will be staff nurses available in the ER so they can help us with IV line, sending lab investigations and with catheterization. But this is a bitter truth that we have shortage of staff. No doubt the staff present in wards does work like they do patient's monitoring, IV medications and follow doctor's instructions (KII- Admin Registrar)

### Lack of adequate resources and quality assurance

HCPs, mainly doctors, and nurses working in the hospital, voiced concerns over the scarcity of resources. All HCPs indicated their workplace as a low-resource setting and described private hospitals as having 'more resources than us'. Despite the disparity in resources, HCPs generally believed they were maximising sepsis management within the limits of what was possible in their unit.

…this is not a private hospital and unit like that. This is civil hospital and we have to face many things. Our surroundings are not as favorable as it seems. We have to struggle a lot and this is the cause of delay in things. But anyways, we are trying our best to manage sepsis cases within our available resources (KII- Registrar Admin)

A majority of the patients present with complications and require intensive monitoring. There are high dependency units (HDUs) and intensive care units (ICUs) in the hospital for critical monitoring of the patients though the shortage of spaces in HDU and ICU is a challenge, as reported by the study participants.

We have monitors available but not according to the patients need. We cannot monitor all the patients and we do it according to the severity of patient's condition. We have only two HDU beds and this is a challenge for us (KII- Senior Registrar)

We have 12 surgical and 12 medicine beds in ICUs altogether in LUMHS for all units. We face constraints of getting ICU beds for critical patients (FGD- HOD)

The OBGYN units have their own set of routines or guidelines that help HCPs organise their practices and influence how and when care is provided. When asked about barriers and enablers in sepsis management, HCPs talked about the lack of awareness of policies that made it difficult to identify and manage sepsis cases. This concern was raised by a few key informants

that a number of HCPs working in the facility are unaware of the hospital policies. Though all the key informants noted the presence of policies and guidelines for sepsis management, only a few (6/16) key informants had detailed knowledge about the policies or guidelines related to sepsis management. The other departments in the hospital example medical ICU, surgical ICU, labour room, emergency room and inpatient wards follow different guidelines for sepsis management. This hinders the care given to patients because no unified system or protocol exists in the facility for sepsis management.

Few people know the correct knowledge of sepsis. People should refresh their knowledge and there should be combined meetings of all units so we have a protocol for CVP lines, high flow oxygen administration and antibiotics. There should be a set vision for this (KII- Senior Registrar)

It was also reported by health administrator of the facility that the workload of HCPs is an impeding factor in sepsis management and causes frustration and burnout among them.

Our doctors are in a hurry to quickly complete their work and go, because they have a lot of burden (KII- Healthcare Administrator)

All HCPs stressed on compromised quality of resources available in the facility. They reported that the quality and efficiency of antibiotics are lacking and there are hurdles in the obtainability of antibiotics. This delays patients' management and the patient care process.

The most important is the below standard antibiotics provided here (FGD- Associate Professor OBGYN)

This is honest truth that the antibiotics we get from outside, from a good company, there is a difference in the quality and efficiency. We are not getting good results with antibiotics as we are supposed to (KII- Senior Registrar)

HCPs also highlighted the constraints faced from the level of patients. The collection and transport of blood samples to laboratories is a complicated process. The patient's samples are transferred to laboratories by the hospital staff at the selected time of the day. If any patient's investigation is required after that fixed set time, it is transferred to laboratory through patients' attendants. Consequently, this delays patients' investigational process.

We have developed a system that in morning, the ward boy will collect samples from each ward, it goes to university hospital which doesn't charge anything. If any sample is missed and sent later, we send them through patient's attendants and they are charged (KII-Health Administrator)

HCPs also deliberated on patient's ability to afford for lab investigations. Most of the patients coming to the facility belong to the low-income class group considering their socioeconomic background. Though LUMHS is a public health facility and a majority of services are provided in the hospital without charge, there are few investigations for which patients are required to pay fees for services for example blood culture and serum lactate tests.

> Our patients are poor and they cannot afford investigations like culture test and serum lactate. They are costly so people are reluctant for these blood tests (KII- Registrar)

> These investigations should be free for patients. Culture bottles are so expensive and people are so poor that they go and throw them away (FGD- Registrar Admin)

### Clinical practice variation
#### Sepsis guidelines and documentation
The interview participants reported that the OBGYN units follow Royal College of Gynaecology (RCOG) guidelines. The RCOG guiding principles provide information about the risk factors of maternal sepsis, the basic vital signs and identification of maternal sepsis, clinical features suggestive of sepsis, investigations to rule out maternal sepsis, and the specific antimicrobial therapy for management.[24] Despite the presence of guidelines in the hospital, the early identification and management of sepsis is a huge struggle.

> MEOWS chart was there in RCOG guidelines and we used to do that, but as you have these FAST-M tools, we didn't use to do this way. We used to do this very haphazardly (KII- Assistant Professor)

The F in the pneumonic of FAST-M denotes fluid resuscitation. This administration of intravenous fluids can be a key intervention for management of sepsis if it is associated with hypotension, however, rapid fluid administration is more complex in pregnant women if there are other coexisting medical problems such as eclampsia. These concerns and delays in fluid administration in the existing system were identified by HCPs. This delay was because of the HCPs anticipated apprehensions and concerns related to complications of fluid therapy as stated:

> In existing practices, we are giving the antibiotics but this fluid therapy sometimes gets delayed as we are concerned about the development of pulmonary edema in septic patients after giving fluids (KII- Registrar)

> Sometimes these gynae people get worried that whether it is sepsis or cardiac issue and whether we should give fluids or not as the patient can have fluid overload (FGD- Assistant Professor- Medicine)

Most of the study participants stated that they are following similar procedures and guidelines as provided in FAST-M bundle care tools. Yet, they identified a lack of documentation in the existing practices.

> We do not follow the step wise procedure and documentation but we follow the same thing as we do respiratory rate, BP, GCS and etc. (KII- Fellow-ICU)

#### Individual care practices and HCP comfort levels
There is a hierarchy of doctors in the hospital from senior to junior level based on their qualifications and experience. The hospital units are managed by Professors who are head of department of the units. The upper category in the hierarchy of doctors comprises all the faculty staff including associate professors and assistant professors, the second upper category in the hierarchy covers registrar doctors, who support postgraduate residents and house officers who come for their internship programme following completion of medical training. These all categories of doctors have diverse job roles for the management of patients as stated:

> We have faculties and we have them on senior level, then we have our Registrars, PGs and HOs, so suppose senior level look for all the patients, do patients rounds and check and advice for the patients. Registrars have their assigned patients' beds. The registrars are assigned according to the number of beds present and occupied. These registrars are accompanied by PGs. Suppose, if any registrar is assigned 12 beds, she gets two PGs who can look after 6-6 beds. So the main people who are on the floor are registrars and PGs who manage patients according to the faculty's advice (KII- Associate Professor)

Within the hospital, it was observed that HCPs' approach to sepsis management was not consistent. Clinical practice variation refers to patients receiving differing care depending on when, where and by whom they are being cared for, despite evidence for best practice. One HCP noted that:

> Some doctors send lactate and culture test and others don't… this may be because of patient's financial affordability. And this variation is also there when we prescribe antibiotics. Every doctor has their own practice (KII- Registrar)

Some nurses voiced concerns about timely management of patients. HCPs reported that patients monitoring gets delayed due to shortage of staff nurses to monitor the patients. There are less senior and skilled nurses in the unit to identify and assess the criticality of the patient. The novice nurses are inexpert to take care of the patients and they also lack skills towards sepsis care.

> Senior nurse makes the schedule and look after the labor room as well as ward because of their competencies. We have new nurses as well but it is obvious

that their understanding and knowledge of the work is less than ours (KII- Staff nurse)

We get senior and competent nurses in the morning shift because there is more work in morning shifts (KII- Senior Registrar)

Unit practice norms, combined with the HCPs' personal comfort, confidence and skills, inform their practices about sepsis management. HCPs also have varying definitions and criteria for which patients are transferred to ICUs and to sort this process uninterrupted, Head of Departments (HODs) from each obstetrics and gynecology unit decide on the eligibility criteria for admission to ICU.

## HCP's perceptions about FAST-M
### Understanding of the FAST-M bundle
HCPs reported that they were informed about FAST-M bundle care tools from their head of departments who are keen to test this intervention in their local setting. Some HCPs had more opportunities to learn about the components of FAST-M bundle, but other HCPs specifically staff nurses did not know about the FAST-M tools. While all doctors reported having a baseline understanding of FAST-M tools and its components including Maternal Early Obstetric Warning System (MEOWS) chart, decision tool and treatment tool, they expressed the need of additional understanding of FAST-M tools before its implementation. All HCPs recommended providing additional education and training sessions to HCPs to address such gaps.

Whatever HCPs are doing, they are doing at their own, they are also trained but they are not very well trained, so training will help them to manage patients well according to the guidelines (KII- OR Doctor)

Healthcare administrators and doctors employed at the hospital displayed their interest in support for implementation of FAST-M intervention, whereas nurses most frequently cited satisfaction with their existing practices.

Our OBGYN doctors are already providing us the charts for monitoring of cesarean deliveries, for baby's monitoring and there are different charts for monitoring. We are already managing our patients well (FGD- Nurse)

Majority of the key informants highlighted positive influences of implementation of FAST-M bundle care tools on existing policies of sepsis management in the hospital as one of them stated:

There is no current guideline followed in the hospital and this has come as a sort of guideline that can be used for sepsis management (KII- OR Doctor)

### Perceptions about significance of FAST-M
HCPs attitudes towards FAST-M implementation were positive and supportive. All HCPs shared positive perceptions about timely sepsis identification and management through classification of patients using MEOWS chart's triggers as red and yellow flags. The use of colours such as red flags and yellow flags indicating cut-off values facilitates HCPs in identifying and categorising patients. HCPs identified colour demonstration in the MEOWs chart as a major enabler in identification of sepsis patients.

Now we know that there is a red and yellow flag, and if patient is in severe sepsis we have to send the samples within an hour and have to give antibiotic and fluids as described in the protocol (KII- Registrar)

It is very easy because of colors we are getting alert on red and yellow flags. This is very easy and understandable (KII- Senior Registrar)

HCPs believed that FAST-M tools improve knowledge of HCPs as the tools include everything related to the identification and management of the patients with maternal sepsis. The flow of the tools was appreciated by HCPs and they also stated that this organised flow of FAST-M tools will save time in sepsis management.

This tool provides specifications about fluid therapy and antibiotics administration with specific time. It has improved our knowledge (KII- Nurse)

HCPs also indicated the significance of FAST-M tools as being initiated by any HCP including the nurse. There is no requirement of a doctor to initiate the bundle care tools. The staff nurses and even the trainee dispensers, who are available in the unit as helpers to staff nurses, can initiate the MEOWs chart for identification of the cases.

The good thing I see in this FAST-M is that even the nurse can start this bundle care (FGD- HOD Gynae)

Generally, most HCPs stated that the FAST-M intervention will help in sharing tasks between HCPs and it will increase the accountability of HCPs to perform their responsibilities

It should be done because from staff till doctor everybody will be responsible for their work and will document each and every thing. We get tired of emphasizing this (KII- ICU Fellow)

One of the KIs emphasised the quality of this tool as being non-invasive. Patients would easily accept this intervention and HCPs would not hesitate to initiate it. It can be easily accepted and implemented.

The intervention that has been introduced, it is totally non-invasive and it is the same work that we do in our daily routine, so we will have no problems in its implementation (KII- ICU Fellow)

All the key informants and focus group participants articulated patients' benefits through FAST-M implementation. They emphasised that the early identification and management of maternal sepsis through the FAST-M tools may decrease patients' length of stay in the hospital,

and eventually decreasing the length of stay would benefit patients in providing physical, economic and psychological advantages. Ultimately, this would help in decreasing maternal morbidities and mortalities in the long run.

> …it will benefit patient that it will help in decreasing the stay of patients and their exposure will be reduced. This will reduce morbidities and mortalities in the long run (KII- Registrar)

### Identifying solutions to the application of FAST-M

Some HCPs were doubtful of the practicality of intervention in the prolonged and continuous implementation due to resource restrictions (eg, quality of available antibiotics, shortage of staffing, shortage of equipment and monitors). The inability to overcome these limitations led to a common attitude that:

> Nothing is sufficient from top to bottom, we try our level best to provide but we do not have monitors, we have hurdles for lab investigations, there are issues of availability of nurses and antibiotics, there are many technical gaps (KII- Registrar Admin)

All respondents suggested that in order to strengthen the significance of FAST-M intervention for early identification of sepsis, the inclusion of the variable of oxygen saturation in the MEOWS chart, with appropriate cut-off values, would be important. This was because pulse oximetry is now available routinely in the unit and may be an important indicator of clinical deterioration. This feedback was consistently given by all HCPs.

> Oxygen saturation is mandatory to include in the MEOWs chart for monitoring of patient (FGD-Assistant Professor- Medicine)

It was informed through HCPs working in the medicine unit that sepsis guidelines followed in their unit include an addition of steroid therapy and inotrope support for sepsis management.

> You should include support because sometimes when we give fluids and antibiotics, but still patient is not maintaining the blood pressure because most of the times septic patients arrives late, so you should include source plus support in S. so both of the things will be included. Because support is the most important (FGD- Assistant Professor- Medicine)

All HCPs agreed over the use of ceftriaxone as first choice of antibiotics in FAST-M treatment bundle based on its cost and availability for patients.

> We give Ceftriaxone straight away as it is freely available. We give 2g Ceftriaxone and for those patients whose culture is sent, we wait for their blood culture reports to change antibiotics accordingly. Otherwise, our patient mostly responds to ceftriaxone (KII-Senior Registrar)

Few participants specified that they use piperacillin/tazobactam and meropenem for management of the confirmed cases of sepsis due to their beneficial results in such patients, yet the patients pay out of pocket for the cost of these antibiotics. Thus, meropenem and piperacillin/tazobactam were proposed as the second choice of antibiotics due to their availability and cost.

> …sometimes when we do not have availability of meropenem so we give ceftriaxone to the patients, which is easily available free of cost for patients (KII-Senior Registrar)

HCPs also suggested involving nursing interns and trainee dispensers who come for their training and work without wages. The involvement of nursing interns and trainee dispensers would reduce the problem of shortage of staffing in the unit and they would be employed to implement the FAST-M intervention without added investment for human resources.

> We get one or two girls from BScN programme, but we can talk to the dean in account and there are many people who can help us with this (FGD- Health Administrator)

The focus group participants identified the need of increasing awareness which is the key to implementation of the FAST-M intervention. The stakeholders emphasised understanding of HCPs about the significance of FAST-M bundle care tools as a key to effective implementation in future. One of the group participants suggested:

> We can make big boards and we can involve everyone and give them awareness. And we can provide examples to them that how it was implemented in past in different setting showing good outcomes (FGD-HOD Gynae)

Moreover, the inclusion of MEOWs charts in patients' Medical Record files of the hospital was emphasised by every group member involved in the discussion.

> We will include MEOWS chart in all patients' files so our doctors can easily record the findings on MEOWS chart which will alert them about patient's condition (FGD- HOD Gynae)

### HCPs readiness for FAST-M implementation

The HCPs readiness towards FAST-M intervention started with the drive of identification of requirements for FAST-M adaptation and concluded with the consensus building of HCPs for its implementation.

### Understanding and identifying gaps

HCPs acknowledged that successful implementation of the FAST-M intervention would require healthcare facility to be well equipped, including both the availability of equipment and trained HCPs. Other key challenges to the successful implementation of FAST-M intervention are related to logistics, including shortage of human

resources and inadequate funds for procuring monitors for assessments, antibiotics and lab investigations. One of the most frequent concerns around FAST-M implementation included the need to train HCPs including doctors, nurses and auxiliary support staff to enable them to set up and sustain the services. Further, study participants suggested that a multidisciplinary approach would be useful to ensure that all professionals including the team of doctors, nurses, administrators from different units for example, medicine, ICUs, labour room, laboratory and operating room are working together for the successful implementation of FAST-M.

> In team, one person should be from administration, to who if we complain related for our hurdles and queries, so he can work on them, one person should be from laboratory, one should be from nursing staff and one should be from doctors, who can take all the things to higher levels and work on them (KII-Registrar Admin)

HCPs argued that there are high costs associated with the implementation of FAST-M intervention. Providers further explained that high costs of laboratory investigations would be a limiting factor as it would cause financial burden to the patients. On the other hand, few health professionals confirmed that costs would not be a major concern if there will be a buy-in from hospital administration for the patient's requirements. HCPs mentioned that the initial investments may be higher for procuring required equipment like monitors and apparatus required for monitoring of patients.

> Ceftriaxone is easily available in our hospital, but we are not sure about its quality. But for the critical patients if we see any red flags, we can arrange their requirements from our donations. In our unit, we are doing this for critical patients (FGD-HOD-Gynae)

### Consensus building for FAST-M implementation

The focus group participants displayed readiness for implementation of FAST-M intervention in their local context by developing consensus on resolutions and approaches to the perceived challenges they could encounter during the implementation. The FGD provided the opportunity to reflect on the anticipated challenges and how they may be able to successfully implement in their setting with the available resources. HCPs decided to implement FAST-M intervention in their setting and they also acknowledged the importance of a training programme for HCPs to implement FAST-M bundle care tools in their setting. It was recognised that the FAST-M protocol comprises similar practices but in an organised and structured way, and was well regarded by all HCPs. They valued the implication of FAST-M intervention as stated:

> We are already doing these all things except documentation so it will be easy to apply. You know the guidelines, you have got an algorithm then it would

be difficult to miss any patient. So it's a very good thing and this can be implemented. We have everything but there should be training and if you give that it would be easy to implement: (FGD- Associate Professor- Medicine)

## DISCUSSION

Our findings revealed several potential facilitators for the uptake of FAST-M intervention. First, the HCPs had highly favourable perceptions regarding the use of FAST-M bundle care tools. The major advantage identified was illustration of coloured codes in the MEOWs chart such as red and yellow flags that assists in categorisation of patients according to severity of their symptoms. The early identification of patients with maternal sepsis through MEOWs chart facilitates timely management of patients using decision and treatment tools.

Evolving morbidity can be difficult to recognise in the obstetric population because of the normal changes in peripartum physiology.[25] Delays in recognition of patient deterioration and initiation of treatment lead to worse outcomes in maternal populations.[25] Early Warning Systems have been used since 1999 in the general patient population to identify clinical deterioration,[26] though the MEOWS has been promoted with the aim to reduce maternal morbidity and mortality, and improve clinical outcomes.[27] The FAST-M intervention comprises different components for the recognition and management of maternal sepsis (online supplemental file 4).

During the development of the FAST-M bundle through a modified Delphi process, oxygen saturation was mostly perceived as of reasonable importance. Though, the feasibility of implementing this element in low-resource settings limited its usefulness due to the non-availability of pulse oximeters at that time in many low-resource settings.[6] However, considering the outbreak of COVID-19 infection and the availability of pulse oximeters at the study site, it was recommended to include oxygen saturation in the MEOWs chart to determine patient's clinical condition. The inclusion of oxygen saturation in the MEOWs chart is considered important based on the existing sepsis management practices of the facility. Moreover, the element of oxygen saturation is a significant indicator in the identification of patients' clinical conditions. Therefore, the supplementary element of oxygen saturation has been added to the bundle care tools prior to its implementation (online supplemental file 6).

The MEOWS chart in the FAST-M intervention tracks physiological parameters and evolving morbidity and once a predetermined threshold reaches, it triggers evaluation by a HCP.[27] The healthcare professional determines further evaluation, treatment, or intervention as necessary through the use of decision tool and treatment bundle.[28] The systematic approach for screening and management of maternal sepsis patients through the

FAST-M intervention supports its implementation in the low-resource setting in Pakistan.

All HCPs acknowledged the FAST-M bundle care tools as easy to use as they do not require any invasive procedures to identify suspected maternal sepsis cases and trigger appropriate actions. Second, the HCPs deliberated about long-term improvement in patient's health outcomes through the use of FAST-M intervention such as the decrease in length of patients' stay at the hospital, and improvement in maternal morbidities and mortalities overall.

Our study findings identified that the shortage of HCPs hindered many aspects of sepsis care delivery, and may be a critical barrier to any intervention. As the hospital provides free of charge care to patients, there is high influx of patients in the facility. This high volume of patients' that increases workload on HCPs and eventually the shortage of healthcare workers is associated with adverse patient outcomes and comprised quality in patient care.[29] Therefore, all the study participants suggested involving nursing interns, trainee dispensers and other available human resources to reduce doctors' and nurses' workload through shared responsibilities and employing a task-sharing approach. The approach of task sharing of specialists with trained non-specialist workers has provided positive outcomes in the improvement of patient care, reduced morbidity and mortality rates, and cost-effectiveness.[29]

Accordingly, a training programme has been planned as part of the implementation of the FAST-M intervention so all HCPs providers have the required knowledge to manage sepsis cases according to the FAST-M approach, making practice uniform across teams in the facility and ensuring the sustainability of FAST-M intervention as a long-term benefit for patients.

The source identification denoted as 'S' in the FAST-M bundle requires a detailed history and examination to identify the infection source along with the targeted further investigations. The training programme will provide an opportunity to improve this aspect, including the significance of taking a detailed history and examination and documenting them. This is very important to provide quality care and to help HCPs to plan a patient's treatment to maintain the continuum of care.[30]

The FAST-M implementation in districts of Malawi provided useful example of effective implementation where champions played a significant role in implementing FAST-M intervention, and their contribution for intervention provided day-to-day oversight of healthcare practitioners' practice.[9] Our study findings suggest that the clinical practice variations among HCPs is a potential major hindering factor in implementation of FAST-M intervention, and yet we decided to select maternal sepsis champions. These champions could potentially standardise the practices for the management of maternal sepsis in all the departments managing such cases. To continue to strengthen the implementation of this intervention, champions will be selected during training programme based on the consensus of HCPs involved in the training of FAST-M intervention.

Moreover, the HCPs were concerned about the compromised quality of available resources such as antibiotics and laboratory investigations which voiced their uncertainty to support FAST-M intervention. They felt that the hospital's environment and the quality of available resources did not support patients' clinical management. It was identified that the hospital system set for laboratory investigations is lengthy and time-consuming.

While the quality of health services within the clinical setting is imperative to provide effective care to the patients.[31] Study findings also suggest that the treatment cost adds to the financial burden of patients and leads to the discontinuation of medical treatment.[32] Thus, the practicability of intervention depends on the facility environment, availability of resources and its affordability for implementation and the readiness of 'healthcare administrators' who are accountable for provision of healthcare supplies. The role of healthcare administrators in upgrading the system is quite significant to avoid barriers to implementation. Hence, the healthcare administrators provided assurance for provision of supplies and resources as a stance to reduce maternal sepsis rate at their healthcare setting and will be fully included in the implementation process, including the training and champion network.

Some specialists raised consideration of broadening the bundle to include more comprehensive sepsis care including consideration of steroid therapy and inotrope support. As part of the adaptation process, this issue was fully discussed with a range of local and international experts from the gynaecology and intensive care fields and it was decided that these aspects would be most appropriate if initiated by specialist doctors, normally in an ICU environment, so would not be suitable for inclusion in the first response bundle. However, the management of patients using steroids would be emphasised during the training programme to delineate its role in the management of COVID-19 as a distinct situation from other bacterial causes of maternal sepsis to ensure rational and evidence based steroid use.

Antibiotics administration is one of the easily available, free of cost and important components of FAST-M treatment bundle for sepsis management. The FAST-M treatment bundle applied in the earlier study conducted in Malawi[9] was therefore of the important. We explored HCPs' views regarding use of antibiotics in their local setting for treatment of maternal sepsis. It was identified that Ceftriaxone is easily available free of cost to patients and it provides positive results in treatment of sepsis. Thus, it was agreed to use ceftriaxone as first choice of antibiotics in FAST-M treatment bundle. Moreover, it was also acknowledged that piperacillin/tazobactam and meropenem are used for treatment of confirmed sepsis cases due to the current understanding of the organisms responsible for maternal sepsis and the antimicrobial resistance patterns. Though patients pay out of pocket

for the cost of these antibiotics. Thus, meropenem and piperacillin/tazobactam were proposed as the second choice of antibiotics due to their availability and cost. The Malawian version of FAST-M treatment bundle was therefore modified for locally appropriate antibiotic guidelines (online supplemental file 6).

The importance of an explicit sepsis care policy was discovered during interviews and FGD to assist in standardising infection regulations in the hospital. It was identified that the FAST-M intervention can serve as a guiding policy to provide evidence-based information to support clinical decision-making. Therefore, a unified system of FAST-M intervention for sepsis care in the facility for maternal patients can serve as a standard tool for maternal sepsis management.

The major strength of this study is the use of CFIR that guided the researchers' focus, starting with observations and documenting from a broad health systems and programme implementation perspective, becoming more specific in the later performed interviews and FGD. Moreover, participation of HCPs from several levels to ask their feedback on the research question, and by interviewing HCPs about their experiences helped in gaining better insights about their practices and perceptions.

The study also has some limitations. First, the study focused only on the perspective of the HCPs who were involved in the management and treatment of maternal sepsis patients; therefore, the sample size was limited and important perspectives from patients and their families could have been missed. Second, the intervention would be implemented in only one study setting in Pakistan at this time. However, it is notable that this site serves a diverse population from the urban and rural areas of province of Sindh. The FAST-M tools were specifically adapted according to the existing sepsis practices of the current study setting. Future studies to explore feasibility of FAST-M bundle would require adaptation prior to implementing in other low-resource settings of Pakistan.

We believe that it is possible to implement the FAST-M intervention in low-resource settings of Pakistan and we recommend several strategies to address the challenges facilities may face in their local context. The hospital, leadership and HCPs require collaboration to work as a multidisciplinary team to advance sepsis management practices and understand its implications. This could be achieved through development and dissemination of FAST-M intervention as a sepsis management guideline in the facility.

The distribution of supportive resources to provide education to all HCPs including doctors, nurses and healthcare administrators about FAST-M tools is required to increase knowledge and awareness of FAST-M bundle. Also, facilities will require selected champions for implementation of the FAST-M intervention.

Overall, bundle care tools have the potential to enhance improvements in sepsis care. However, the implementation challenges posed by these bundles should be examined, especially in low-resource settings, where facilities and services have not yet flourished.

We identified facilitators and barriers for implementation of this intervention from only one of the facilities in Pakistan selected as our study site. Future research is needed to understand how implementation of this adapted FAST-M intervention works when implemented as part of care, and to rigorously evaluate its effectiveness and key implementation outcomes such as the sustainability of the intervention.

## CONCLUSION

The FAST-M maternal sepsis bundle has the potential to be used as an integrated strategy for early recognition and management of maternal sepsis in low-resource health settings in Pakistan. We found several barriers and facilitators for its implementation and suggested key adaptations to the intervention which we perceive will help address these barriers.

Based on this formative research, the FAST-M tools and implementation approach in their adapted format will be implemented in the selected health facility and mixed-methods research conducted to assess the feasibility of implementing these adapted tools as part of the healthcare system in Pakistan.

**Acknowledgements** We would also like to acknowledge Dr. Sadia Shakoor (Associate Professor Section of Microbiology, Department of Pathology and Laboratory Medicine, Aga Khan University Hospital) in providing information and recent statistics related to use of antibiotics in Pakistan. We would like to acknowledge health officials and individuals: Dr. Anna Blennerhassett (for her expert contributions to the project), Dr. Zulfiqar Shah, Dr. Madiha Shah, Dr. Imran Karim Shaikh, Dr. Mumtaz Lakho, Dr. Fouzia Sheikh, Mr. Sattar Jatoi, Dr. Nargis and Dr. Sabrina (for providing assistance, contact numbers and permissions for data collection at field sites).

**Contributors** SIA, DL, RB and LS conceptualised the design of the study and creation of data collection tools. RS, RR and SK assisted in data collection from field site. SIA, RB, BMHK and GKR managed data collection and interpretation. SIA and BMHK carried out the analysis and wrote the initial manuscript. All authors provided input during the interpretation of the data and revising of the manuscript. DL, AC, RB, JC and CLD provided feedback on the first draft. SIA and BMHK edited and wrote the final draft. The authors read and approved the final manuscript. SIA takes responsibility for the overall content as the guarantor.

**Funding** This research study is funded by the National Institute for Health Research (NIHR). Award Ref: NIHR300808 Host: University of Liverpool. The views expressed in this paper are those of the authors and not necessarily those of the NIHR or the Department of Health and Social Care.

**Disclaimer** The views expressed in this paper are those of the authors and not necessarily those of the NIHR or the Department of Health and Social Care.

**Competing interests** None declared.

**Patient and public involvement** Patients and/or the public were not involved in the design, or conduct, or reporting, or dissemination plans of this research.

**Patient consent for publication** Not applicable.

**Ethics approval** Ethical approval for the study was provided by LUMHS hospital (REC/-886, 4-87), Aga Khan University Ethical Review Committee (2019-2061-7102) and National Bioethics Committee (515/20/). Participants gave informed consent to participate in the study before taking part.

**Provenance and peer review** Not commissioned; externally peer reviewed.

**Data availability statement** All data relevant to the study are included in the article or uploaded as online supplemental information.

**ORCID iDs**
Sheikh Irfan Ahmed http://orcid.org/0000-0002-8391-8559
Catherine Louise Dunlop http://orcid.org/0000-0002-4792-9496

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
