## [Reviewer comments · BMJ Open]

ARTICLE DETAILS

TITLE (PROVISIONAL)	Adapting the FAST-M maternal sepsis intervention for implementation in Pakistan: A qualitative exploratory study
AUTHORS	Ahmed, Sheikh; Khowaja, Bakhtawar; Barolia, Rubina; Sikandar, Raheel; Rind, Kubra; Khan, Sehrish; Rani, Raheela; Cheshire, James; Dunlop, Catherine; Coomarasamy, Arri; Sheikh, Lumaan; Lissauer, David

VERSION 1 – REVIEW

REVIEWER	Shields, Andrea University of Connecticut, Obstetrics & Gynecology
REVIEW RETURNED	07-Dec-2021

GENERAL COMMENTS	Authors do a nice job of describing their study question, and methodology is appropriate. How were participants selected for participant in focus groups or as key informants (e.g. sampling of convenience)? Did any participants approach refuse to participate? Table 2 should have expanded demographics about the participants - e.g., year on job, titles, roles in the hospital, gender, and timeframe of interviews. Were the research questions pilot tested? Were any repeat interviews carried out, and if so, why? Discuss data saturation. Please comment if participants provided feedback on the data/themes. In discussion section, I recommend authors discuss why patients were not included in the focus groups. Conclusions are appropriate.
--

REVIEWER	Turner, Michael UCD Centre for Human Reproduction, University College Dublin
REVIEW RETURNED	10-Jan-2022

GENERAL COMMENTS	The subject matter is a critical one in contemporary obstetrics worldwide and the paper is very well written. The authors commitment to their work is commendable. In the Abstract, the date of the study should be stated and information provided about the study setting. "CFIR" should be spelt out. A major weakness of the study is that the number of participants is small, particularly given that the 28 included were multidisciplinary. The selection of participants was also open to bias especially as those interested in the subject of maternal sepsis may have been more likely to participate. A detailed audit of maternal sepsis in the hospital would be informative. What were the causes? Which cases were more likely
---

	to result in maternal mortality? Were the women delivered in the hospital or elsewhere? Was FAST M implemented in the cases of sepsis, for example, did they receive appropriate antibiotic treatment? Were there delays in implementation? A further weakness of the study is that it may not be applicable in other settings either within or outside Pakistan and thus there are limitations on the learning points for readers of the BMJ Open. There should be some discussion about the different early warning systems and their advantages and disadvantages in assessing maternal infection and sepsis. Why choose FAST M over others? It is also notable that the staff nurses working within the hospital did not know about FAST M. This highlights a challenge with clinical guidelines universally, that is, inadequate dissemination and implementation following development. There are too many variables in the intervention itself and in the “Tailored” implementation to draw any conclusions from this study. The authors should however persist with their interest in this important subject but I would recommend more limited and focused clinical audits initially.
--	--

VERSION 1 – AUTHOR RESPONSE

Responses from reviewers

	Reviewer #1: Dr. Andrea Shields, University of Connecticut Comments to the Author: Authors do a nice job of describing their study question, and methodology is appropriate.	Thank you very much for providing your expert feedback. We have incorporated your suggestions in the revised manuscript and tried to respond to your queries below. Hope this clarifies your concerns.
1.	How were participants selected for participant in focus groups or as key informants (e.g. sampling of convenience?)?	The participants’ selection for the study has been explained in lines 220-223
2.	Did any participants approach refuse to participate?	Line 223-224 “All the participants who were approached by the study team agreed to participate in the study” has been added.
3.	Table 2 should have expanded demographics about the participants - e.g., year on job, titles, roles in the hospital, gender, and timeframe of interviews.	The tables (3 &4) have been included on page no:16 to display participants’ demographics

4.	Were the research questions pilot tested?	Lines 238-240 have been added to explain the pilot testing of the interview guides.
5.	Were any repeat interviews carried out, and if so, why?	No, we were able to reach the point of data saturation by conducting 16 Key informant interviews and 1 Focus group discussion.
6.	Discuss data saturation.	Lines 262-266 describe data saturation
7.	Please comment if participants provided feedback on the data/themes.	Lines 283-287 describe the process that was undertaken to obtain participants' feedback for confirmation of findings
8.	In discussion section, I recommend authors discuss why patients were not included in the focus groups.	This study aims to understand the existing sepsis management practices and behaviours to adapt the FAST-M bundle care tools in the local context (Lines 158-160). Based on the study objective, we collected data from healthcare practitioners and health officials only to answer our research questions.
9.	Conclusions are appropriate.	

	Reviewer: 2 Dr. Michael Turner, UCD Centre for Human Reproduction Comments to the Author: The subject matter is a critical one in contemporary obstetrics worldwide and the paper is very well written. The authors commitment to their work is commendable.	Thank you for taking out time to review this paper and providing very useful feedback. We have incorporated your suggestions in the revised manuscript and tried to respond to your queries below. Hope this helps to clarify your concerns.
1.	In the Abstract, the date of the study should be stated and information provided about the study setting. "CFIR" should be spelt out.	We have added the date of study and CFIR framework in lines 61 & 63
2.	A major weakness of the study is that the number of participants is small, particularly given that the 28 included were multidisciplinary. The selection of participants was also open to bias especially as those interested in the subject of maternal sepsis may have been more likely to participate.	As the objective of this qualitative study was to adapt the FAST-M intervention according to the existing sepsis practices in the study setting, we included only healthcare providers, administrators, and stakeholders who were involved in the management and treatment of maternal sepsis. This has been added as a study limitation, lines 759-762

3.	A detailed audit of maternal sepsis in the hospital would be informative. What were the causes? Which cases were more likely to result in maternal mortality? Were the women delivered in the hospital or elsewhere? Was FAST M implemented in the cases of sepsis, for example, did they receive appropriate antibiotic treatment? Were there delays in implementation?	Lines 294-297 explained “A baseline facility audit was conducted to identify the availability of resources in the facility and assisted the study team to plan a practical approach for the implementation of the intervention (audit form has been provided as a supplemental file-5). Lines 211 to 216 provide the information extracted from the baseline survey form. The detailed findings will be recorded elsewhere for before-after implementation outcomes.
4.	A further weakness of the study is that it may not be applicable in other settings either within or outside Pakistan and thus there are limitations on the learning points for readers of the BMJ Open.	Any new intervention requires adaptation and feasibility testing before its implementation. We, therefore, adapted the FAST-M intervention in context of our particular study setting. We do however feel both the process and findings will be of interest to others working in maternity settings with limited resources in other LMICs.
5.	There should be some discussion about the different early warning systems and their advantages and disadvantages in assessing maternal infection and sepsis. Why choose FAST M over others?	Lines 148-154, 650-657, 660-674 discuss different early warning systems and provide rationale for using the FAST-M intervention.
6.	It is also notable that the staff nurses working within the hospital did not know about FAST M. This highlights a challenge with clinical guidelines universally, that is, inadequate dissemination and implementation following development.	Yes, we agree that this is a major challenge universally. We observed that the number of staff nurses working in the facility was very limited which brings another challenge to the communication gap between doctors and nurses and the lack of information sharing. The role of doctors is very prominent in the study setting, though the importance of nurses in patient care could not be ignored as this intervention was developed and conceptualized to be initiated by any healthcare provider. Therefore, we involved nurses to have their views and opinions on the adaptation of this intervention. We have also planned to involve staff nurses in training sessions to have their active role and participation in the implementation. They will also be trained on the use of bundle care tools. These training sessions would support the

		dissemination of information and awareness-raising among nurses. Reference # 19 is the protocol for “Evaluation of the feasibility of the FAST-M maternal sepsis intervention in Pakistan”
7.	There are too many variables in the intervention itself and in the “Tailored” implementation to draw any conclusions from this study.	We recognize that it is a “complex” intervention with multiple, interacting components. We acknowledge that we will therefore not be able to isolate the effects of any specific component. However, this is not the purpose of the study as we seek to understand the complex intervention in its entirety, and within the health system where it will be implemented. FAST-M care bundle was developed using a modified Delphi process which engaged a wide range of healthcare practitioners of LMIC’s. The intervention has already been validated in Malawi, we have used the same tools (implemented in Malawi) to adapt in our local settings.
8.	The authors should however persist with their interest in this important subject but I would recommend more limited and focused clinical audits initially.	We have recorded the baseline audit details elsewhere and those would be published separately. This paper is based only on the qualitative findings gained from HCPs feedback.

VERSION 2 – REVIEW

REVIEWER	Shields, Andrea University of Connecticut, Obstetrics & Gynecology
REVIEW RETURNED	07-Aug-2022

GENERAL COMMENTS	I commend the authors for improving the quality of this interesting manuscript through their revisions. They have adequately answered the peer reviewer questions.
--